# Prolonged Diagnostic Interval Leads to High Lymphoma Related Mortality in a Prospective Cohort of People with HIV Undergoing Fine Needle Aspiration

**DOI:** 10.3390/cancers17061005

**Published:** 2025-03-17

**Authors:** Samantha L. Vogt, Khuthadzo Hlongwane, Arshia Arora, Kennedy Otwombe, Deshan Chetty, Rebecca H. Berhanu, Ziyaad Waja, Wendy Stevens, Tanvier Omar, Neil A. Martinson, Richard F. Ambinder, Rena R. Xian

**Affiliations:** 1Department of Medicine, Johns Hopkins School of Medicine, Baltimore, MD 21205, USA; aarora17@jhmi.edu (A.A.); rambind1@jhmi.edu (R.F.A.); 2Department of Oncology, Sidney Kimmel Comprehensive Cancer Center, Johns Hopkins School of Medicine, Baltimore, MD 21205, USA; 3Perinatal HIV Research Unit (PHRU), Faculty of Health Sciences, University of the Witwatersrand, Johannesburg 2193, South Africa; hlongwanek@phru.co.za (K.H.); otwombek@phru.co.za (K.O.); chettyd@phru.co.za (D.C.); wajaz@phru.co.za (Z.W.); 4School of Public Health, Faculty of Health Sciences, University of the Witwatersrand, Johannesburg 2193, South Africa; 5Department of Medicine, Division of Infectious Diseases, Vanderbilt University Medical Centre, Nashville, TN 37232, USA; rebecca.h.berhanu@vumc.org; 6Wits Diagnostic Innovation Hub, Faculty of Health Sciences, University of the Witwatersrand, Johannesburg 2193, South Africa; wendy.stevens@wits.ac.za; 7Division of Anatomical Pathology, National Health Laboratory Service, Faculty of Health Sciences, University of the Witwatersrand, Johannesburg 2193, South Africa; tanvier.omar@nhls.ac.za; 8Department of Pathology, Johns Hopkins School of Medicine, Baltimore, MD 21205, USA; rxian1@jhmi.edu

**Keywords:** HIV, fine needle aspiration, lymphoma, tuberculosis, mortality

## Abstract

HIV is associated with an increased risk of aggressive lymphomas. In low-resource settings, fine needle aspiration (FNA) is an important diagnostic modality used in the evaluation of lymphadenopathy to help differentiate between tuberculosis (TB) and lymphoma. In this prospective, observational study, we set out to better understand the diagnostic triage through the FNA clinic for PWH presenting with lymphadenopathy in Soweto, South Africa. One hundred and forty-six PWH were recruited and prospectively followed up for up to 8 months. We report on the overall pathology, time to lymphoma diagnosis, and 8-month mortality. In our cohort, TB was the most common diagnosis found on FNA, followed by lymphoma. However, lymphoma was the leading contributor to death. Among those with lymphoma, roughly half of participants died prior to chemotherapy initiation. Strategies to improve the diagnosis of lymphoma in South Africa are needed.

## 1. Introduction

In people with HIV (PWH), there is a broad differential diagnosis for lymphadenopathy that spans benign processes, infectious diseases, and malignant conditions. Fine needle aspiration (FNA) is an important diagnostic modality used in the evaluation of lymphadenopathy in PWH in low-resource settings to help differentiate among these diagnoses [1,2,3,4]. The overlap between constitutional symptoms, such as fever, night sweats, and weight loss with lymphadenopathy in PWH, may reflect an infectious process or an aggressive lymphoma [5]. In South Africa, a country with roughly 7.5 million adults living with HIV as of 2023 [6] and one of the highest rates of TB co-infection [7], distinguishing between tuberculosis (TB) and other causes of lymphadenopathy is particularly important. Due to the high prevalence of TB in this region, FNA specimens undergo routine testing with GeneXpert Ultra, a molecular diagnostic test, and a TB culture, complementing the standard cytologic analysis.

As HIV is associated with an increased incidence of aggressive lymphomas [8,9,10,11] and lymphadenopathy is common at the time of presentation, FNA represents an important diagnostic tool in PWH in South Africa. However, FNA alone is not diagnostic and cytology findings that are suggestive of lymphoma need to be confirmed with a tissue biopsy. A retrospective review conducted by our group previously showed that over one-third of PWH with cytology suggestive of lymphoma on FNA did not undergo a confirmatory biopsy [1]. Given the retrospective nature of the prior study, the time from the initial presentation to the healthcare system until a lymphoma diagnosis was not known, and the contributing factors leading to loss-to-follow-up were poorly understood. In the current study, we set out to better understand the diagnostic triage through the FNA clinic for PWH presenting with lymphadenopathy in Soweto, South Africa.

## 2. Materials and Methods

We conducted a prospective, observational study of PWH undergoing lymph node FNA at Chris Hani Baragwanath Academic Hospital in Soweto, South Africa, a large tertiary referral hospital in the South African public healthcare sector. Geographically, Soweto is in the southwestern part of Johannesburg and has an estimated population of roughly 3 million, predominately Black, individuals. The public healthcare system in South Africa is a tiered system consisting of primary healthcare clinics (PHCs), community healthcare centers (CHCs), and district, regional, and tertiary hospitals. Patients presenting to the PHCs or CHCs who require a higher level of care, such as specialized diagnostic services like an FNA, are referred to specialty clinics located at the referral hospitals. Chris Hani Baragwanath Academic Hospital has an established FNA clinic that accepts referrals from the 22 PHCs and six CHCs within Soweto.

The eligibility criteria for the observational study included age ≥ 18 years, a documented HIV infection, and the willingness and ability to provide consent in either English, Zulu, or Sesotho. Pregnant women, prisoners, and individuals with a prior history of malignancy were excluded. This study was approved by the Johns Hopkins School of Medicine Institutional Review Board (IRB00165673) and the University of the Witwatersrand Human Research Ethics Committee (180216).

Between September 2021 and December 2022, 365 patients undergoing FNA were referred to our study team. Two hundred and nineteen patients were excluded, and 146 participants were enrolled in the observational cohort (Figure 1). At the time of enrollment, demographics, including sex assigned at birth and self-identified race, and a brief clinical history, including exposure to antiretrovirals (ARVs) and TB history, were collected, and an Eastern Cooperative Oncology Group (ECOG) performance status (PS) assessment was performed. Participants were also asked when their symptoms began and when and where (clinic/hospital, pharmacy, or traditional healer) they sought care for their symptoms. Consent for the medical and laboratory record review was obtained.

Study participants were prospectively followed up for at least 8 months after the FNA procedure with serial follow-up visits scheduled at 2, 6, 14, 24, and 32 weeks. At each follow-up visit, participants were asked whether they had received the results of the FNA and whether they had undergone any further work-up, including a referral for biopsy (if indicated), and a brief clinical history, symptom assessment, and physical exam were performed. At the study follow-up visits, participants who had not received their FNA results were provided a printed copy of the result, provided that the result was available in the NHLS system. All participants were encouraged to return to their clinic for follow-up to receive/discuss their FNA results and any ongoing symptoms that they were experiencing. For participants diagnosed with lymphoma, a medical record review of the hematology files was performed. Abstracted data included the date of chemotherapy initiation and survival for up to 2 years. Participants without a CD4 count in the National Health Laboratory Service (NHLS) laboratory database in the 6 months preceding enrollment were consented to have a CD4 count collected during their next follow-up appointment. Participants were reimbursed 150 South African Rand (~$10.00) for each visit to cover their time and travel costs. Phone interviews were allowed for participants who were unable to return for follow-up visits. Participants’ next of kins were contacted if the study team was unable to reach the participant. In case of participant death, the family was asked whether the participant was hospitalized at the time of death. When death occurred in-hospital, records were obtained, and the NHLS laboratory database was reviewed to better understand the cause of death.

### 2.1. FNA Testing

FNA cytology specimens were processed through the NHLS. As part of routine clinical care, two FNA passes are typically obtained. One pass is sent for TB testing with GeneXpert Ultra and a TB culture, and the other pass is sent to cytology for review by a pathologist. The FNA results were categorized based on the cytology review and TB testing as follows: TB, lymphoma, reactive adenopathy, benign/cyst, other malignancy, inflammation/abscess, or inadequate. Participants in the lymphoma category included any participant who had a biopsy-proven lymphoma during study follow-up, regardless of initial FNA cytology, and any participant who had cytology suggestive of lymphoma over the course of the study, regardless of whether they had a biopsy to confirm the diagnosis. For a participant to be included in the TB category, any of the following criteria needed to be met: (1) TB detected by GeneXpert; (2) positive TB culture; or (3) cytopathology consistent with TB.

### 2.2. Statistical Analysis

Participants were prospectively followed up until a diagnosis of lymphoma was established. A secondary outcome was 8-month survival. Patient and healthcare provider intervals were calculated for participants in the lymphoma category [12,13,14]. The patient interval was defined as the time from symptom onset until the participant sought care within the healthcare system at either a clinic or a hospital. For participants who reported presentation to the healthcare center prior to referral to the FNA clinic, data triangulation through a review of the NHLS laboratory database was performed to help improve the validity of the self-reported history. Histories that were not consistent with the laboratory database were not included in the analysis. The healthcare provider interval was defined as the time from participant presentation to the healthcare system until a lymphoma diagnosis was confirmed on a biopsy (date of the biopsy result). The date from the biopsy confirmation to the initiation of chemotherapy and overall survival was also calculated based on abstracted data from the hematology clinic. The healthcare provider interval for those participants who died prior to a lymphoma diagnosis represents the time from presentation to the healthcare system until the time of death.

Percentages and frequencies were used to present the categorical data. Medians and interquartile ranges described continuous measures. A Chi-square test was used to compare the categorical variables and Kruskal–Wallis was used for continuous comparison. Wealth quintiles were calculated as previously described in the Measuring Equity with Nationally Representative Wealth Quintile and Steps in Constructing the New Demographic and Health Surveys Wealth Index [15,16].

## 3. Results

One hundred and forty-six PWH undergoing FNA were enrolled, including 70 males (48%) and 76 females (52%; Table 1). The median age was 40 years (interquartile range: IQR 33–49) and the median CD4 count was 216 cells/μL (IQR 102–408; *n* = 138). Most participants were on ART (113; 77%) for >1 year (70; 63%); however, only 39 participants (27%) had an undetectable HIV viral load. The median HIV viral load for those with a detectable level was 2.66 log copies/mL (IQR 2.8–4.86). The majority of participants had an ECOG performance status of ≤1 (116; 79%). At the time of the FNA procedure, 56 participants (38%) reported a prior history of TB, and 21 participants (14%) were on TB treatment.

The findings from the FNA were as follows: TB in 61 PWH (42%), lymphoma in 18 (12%), reactive adenopathy in 17 (12%), benign or cyst in 16 (11%), inadequate in 15 (9%), abscess or inflammation in 12 (8%), other malignancy in 8 (6%), and TB and lymphoma in 1 (1%; FNA cytology was suggestive of lymphoma and GeneXpert Ultra was positive for TB). An additional two participants were ultimately diagnosed with lymphoma during the study follow-up but had initial FNAs that showed reactive adenopathy. One participant had a repeat FNA that was suggestive of lymphoma before proceeding to a confirmatory biopsy, and the other proceeded directly to a lymph node biopsy.

There was no difference in the median age, gender, or wealth quintile between those with and without lymphoma. There was also no difference in the median CD4 count or exposure to ARTs in participants with lymphoma compared with those without lymphoma. However, participants with lymphoma reported a longer duration of antiretroviral use (>1 year ART: 83% in lymphoma participants vs 59% non-lymphoma participants; *p* = 0.0463). Self-reported prior history of TB and current TB treatment were similar between both groups.

Among the 21 participants with either suspected or confirmed lymphoma, only 11 participants (52%) survived long enough to have their lymphoma confirmed on a biopsy (Table 2). Eight participants (38%) died prior to lymphoma diagnosis, including 3 participants who had a diagnostic biopsy performed but died prior to the results being released. The remaining 2 participants (10%) were alive and still awaiting a biopsy confirmation at the end of the 8-month study follow-up period. For both participants, several attempts were made to contact them and their next of kin to make them aware of the FNA result and encourage them to follow-up at their local clinic. The FNA result and any additional diagnostic procedures performed, including the results of the diagnostic biopsy, are listed in Table 2.

Among the 21 participants included in the lymphoma category, the patient interval was interpretable for 15/21 (71%), including 7/11 (64%) participants who survived until a lymphoma diagnosis, 7/8 (88%) participants who died prior to a biopsy confirmation of lymphoma, and 1/2 (50%) participants who was still awaiting a biopsy confirmation when the study follow-up ended (Table 3). The median patient interval was 30 days (IQR 11–71 days). There was no difference in the patient interval between those diagnosed with lymphoma and those who died prior to a lymphoma diagnosis. For the participants who survived until a lymphoma diagnosis, the majority reported seeking care at a clinic or hospital after symptom onset. This contrasts with those who died prior to a lymphoma diagnosis. Among these eight participants, presentation to a traditional healer or pharmacy after symptom onset was more common.

The median healthcare provider interval was 85 days (IQR 55–134) for those who were diagnosed with lymphoma, which was the time from presentation to a healthcare provider to a biopsy-proven lymphoma diagnosis. Of note, the healthcare provider interval could only be confirmed for 10 of the 11 lymphoma participants, as the date of the biopsy result was unknown for one participant. The median healthcare provider interval for those who died prior to diagnosis was 97 days (IQR 33–134), defined as the time from presentation to a healthcare provider to death. Figure 2 shows the components of the healthcare provider interval.

Among the 11 participants who survived to have a lymphoma diagnosis confirmed (Figure 3), 9 were able to initiate chemotherapy. Two participants died between 5 and 14 days after biopsy confirmation and were not treated. With a median follow-up of 398 days, six of the nine participants who started on chemotherapy (67%) were alive.

### Contributors to Death

Twenty-three participants died (16%) during the 8-month study follow-up period and two participants (1%) were lost to follow-up. The FNA cytology results for the participants who died were as follows: lymphoma in 10/23 (43%), TB in 7/23 (30%), other malignancy in 3/23 (13%), other infection in 2/23 (9%), and unknown cause in 1/23 (4%). In fact, 48% (10/21) of the patients with lymphoma died during the follow-up period, as compared with 11% (7/61) of patients with TB or 37% (3/8) of patients with other malignancies. Nine of the ten lymphoma-related deaths occurred either during the work-up for lymphoma or prior to starting chemotherapy; the one remaining death occurred within two months of a lymphoma diagnosis. Markers of HIV control (CD4 and HIV viral load), FNA result, co-morbidities, site of death (at home or in hospital), and time to death from FNA procedure are shown in Appendix A, Table A1.

## 4. Discussion

In this prospective, observational study, we report on the devastating consequences of inefficient diagnostic triage for aggressive lymphomas in PWH. Twenty-one (14%) PWH undergoing FNA had findings that were suggestive of lymphoma. Among these, almost half of these patients died prior to receiving chemotherapy and two (10%) were still pending a biopsy at 8 months. The high mortality rate is consistent with our prior retrospective study conducted in the larger Johannesburg region [1]. This finding is also consistent with the results of both retrospective [17,18] and prospective [19] Hodgkin lymphoma cohorts in South Africa, where a high mortality rate for PWH prior to chemotherapy was reported.

This study was distinctive insofar as we enrolled participants prior to a biopsy-confirmed lymphoma diagnosis. To further support our categorization of lymphoma beyond FNA cytology alone, three participants had biopsies performed antemortem but died prior to the biopsy results. We also note a prior study conducted at Chris Hani Baragwanath Academic Hospital showed a high level of concordance between an FNA suggestive of lymphoma and the diagnosis of lymphoma on a subsequent biopsy [20].

A particular focus in the current study was to define the healthcare provider interval. We report a median healthcare provider interval of 85 days. As mentioned above, many participants did not survive for the duration of this interval. For those who died prior to diagnosis, the median healthcare provider interval was 97 days. Given that patients died prior to lymphoma diagnosis, the 97 days underestimates the true healthcare provider interval but also suggests that there was adequate time from the first presentation to the healthcare system to establish a diagnosis. To provide further context for the length of this interval, a prior study from Cape Town, South Africa found that when the healthcare provider interval exceeds 6 weeks, patients were more likely to be diagnosed with late-stage disease [12].

In trying to understand the main contributors to a delayed lymphoma diagnosis through the FNA clinic, several themes emerged. First, during the patient interval, a higher percentage of participants who died prior to a lymphoma diagnosis reported seeking care at a traditional healer or pharmacy at the time of symptom onset. This observation is based on a small number of participants, but the idea that the use of traditional medicine can contribute to a delayed cancer diagnosis has also been reported in Tanzania [21] and Ethiopia [22]. A better understanding of the reasons behind the first presentation to a traditional healer or self-medication at the pharmacy could help to inform on potential community outreach and education campaigns to improve cancer diagnosis.

Next, during the healthcare provider interval, the turnaround time for the FNA result was roughly two weeks. Given the high number of participants who dropped off after the FNA procedure due to death or the lack of a referral for biopsy, this interval is of particular importance. Per discussions with participants during their scheduled follow-up visits, they relayed that they were instructed to follow-up at their local clinic one week after the FNA procedure to discuss the results. Unfortunately, at one week, only the GeneXpert testing for TB would be available. Given that TB was the predominant finding on FNA in 61 participants (42%), this strategy appears appropriate for many participants with TB. The lower mortality rate associated with participants with an FNA diagnostic of TB (7/61; 11%) highlights the success of this follow-up strategy. Unfortunately, the 1-week follow-up necessitates multiple visits for a participant to obtain the complete results of the FNA procedure and could create an unnecessary delay and a barrier to biopsy referral. As malignancies accounted for the majority (13/23) of the deaths in this cohort, and most of these were lymphomas (10/13), the importance of the FNA cytology result might be lacking and physician awareness of the increasing burden of lymphomas and malignancies in PWH might be inadequate. This concept of provider bias is supported by the results of a Peruvian study that noted that physicians tended to over-diagnose TB and under-diagnose malignancy in patients presenting with lymphadenopathy, regardless of HIV status [23]. As the empiric treatment of TB in patients with lymphoma has been well documented in sub-Saharan Africa, [12,17,24,25] educational programs detailing the increasing incidence of malignancies within PWH could be beneficial.

To improve the diagnostic triage for lymphoma, one can envision several approaches. With more trained cytopathologists, it should be possible to reduce the time required for cytologic diagnoses, such that TB and cytology results could be provided together at the one-week follow-up visit. The practicality of such an approach in the public healthcare sector in South Africa is unknown and could take years to implement. Alternatively, a negative GeneXpert result for TB in an FNA specimen might prompt an upfront core biopsy. This approach has been explored in Cape Town, South Africa, where investigators evaluated 130 patients over 2 years and showed a significant decrease in the time from clinic presentation to a diagnostic biopsy [26]. Additionally, the addition of upfront flow cytometry and/or immunohistochemistry to the FNA procedure could be considered. This strategy would come at a significant cost to the healthcare system and would not eliminate the need for a confirmatory biopsy in many cases [27,28]. A more practical approach could include the addition of a care navigator to facilitate the linkage between the FNA and the diagnostic biopsy. Finally, the incorporation of molecular diagnostics for lymphoma might facilitate a more rapid diagnosis [29,30]. We have been exploring the possible role of the detection of clonal immunoglobulin DNA for diagnosis in a prospective study.

Our study has several limitations. First, the time to diagnosis included both the patient interval and the healthcare provider interval. Both intervals relied on patient reports and may be influenced by recall bias. To limit the effect of recall bias, we utilized an electronic laboratory database, an objective data source, to help corroborate the dates reported by participants when possible. Additionally, the date of symptom onset and presentation to the healthcare system were uniformly collected at time of enrollment for all the study participants to decrease the recall period. Secondly, we report on the prevalence of lymphoma in PWH attending an FNA clinic with lymphadenopathy; however, we excluded several groups of individuals, including prisoners, pregnant women, and PWH with a prior history of malignancy. For those with a prior history of malignancy, FNA is a useful tool to assist with the staging of various tumor types and evaluating for relapsed disease when the underlying cancer diagnosis is already known and referral to subspecialty care has already occurred. In the current study, our focus was on FNA as the first diagnostic procedure, not its utility in staging or detecting relapsed disease; this was the rationale for excluding those with a prior history of malignancy. These exclusions could influence the true prevalence of aggressive lymphoma in PWH, although the number of potential participants who fell into any of these three groups was small (*n* = 22). Additionally, the overall sample size was limited due to enrollment at only one hospital in comparison with the large number of FNAs that are performed in the greater Johannesburg area. From our retrospective study, we reviewed over 500 FNAs performed in PWH over 3 months [1]. In the current study, we enrolled 146 participants over 16 months. Despite the modest sample size, the cytopathologic findings were consistent with our previous report and the high mortality for undiagnosed lymphoma was comparable between the two studies. Post-mortem examinations to better ascertain the cause of death would enhance future studies. As this study was designed as an observational study, and a relatively modest number of participants (*n* = 21) were included in the lymphoma category, our results lack the power to detect differences between those with lymphoma and those without, and those who survived to a lymphoma diagnosis and those who died. A larger follow-up study could be proposed to identify the factors associated with a lymphoma diagnosis.

## 5. Conclusions

FNA is an important screening modality in this high HIV and TB burden region. However, the linkage between a suspicious FNA and diagnostic biopsy appears inadequate, leading to a high lymphoma-related mortality in PWH. The success in molecular GeneXpert testing for TB at the FNA clinics demonstrates that rapid access to results is effective in ensuring patients are appropriately triaged for TB treatment. However, the slower turnaround time for cytology results markedly diminishes the potential benefits for those with lymphoma. Pragmatic solutions, including additional testing, such as molecular diagnostics for lymphoma, to decrease the time to diagnosis and improve patient navigation for PWH and aggressive lymphomas are needed. As these lymphomas are treated with curative-intent chemotherapy, an earlier diagnosis should result in lives saved.

## Figures and Tables

**Figure 1 cancers-17-01005-f001:**
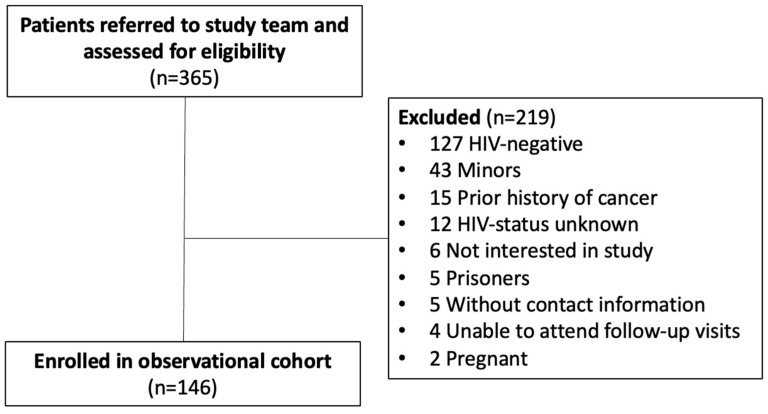
Referral and enrollment flow chart.

**Figure 2 cancers-17-01005-f002:**
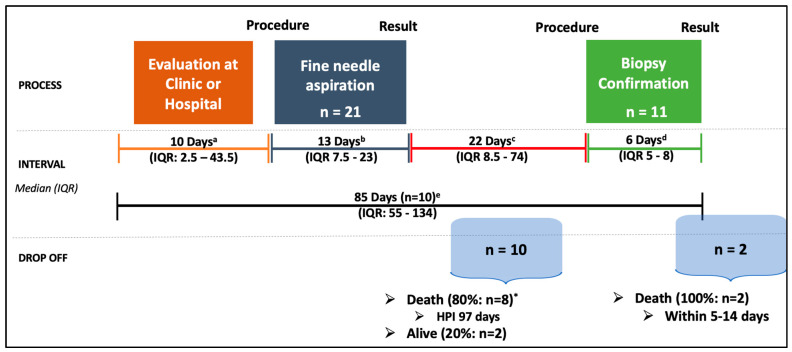
Healthcare provider interval. Healthcare provider interval divided into segments: ^a^ Interval between presentation to the healthcare provider and FNA as obtained from the participant report; ^b^ FNA turnaround time; ^c^ Interval between the FNA result and biopsy; ^d^ Biopsy turnaround time (known for 10 participants); ^e^ Healthcare provider interval for the 10 participants who completed this interval. * Five of the eight participants had a biopsy performed; biopsy was non-diagnostic (*n* = 2); biopsy resulted after participant death (*n* = 3). Abbreviations: IQR: interquartile range; HPI: healthcare provider interval.

**Figure 3 cancers-17-01005-f003:**
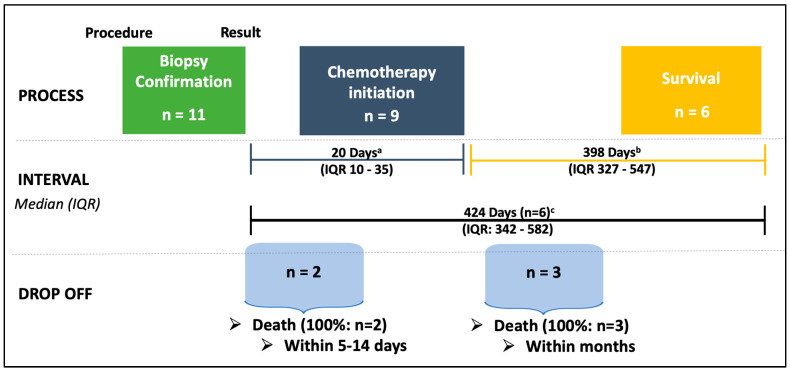
Hematology referral and treatment interval. Hematology referral and treatment interval divided into segments: ^a^ Interval from a lymphoma diagnosis to chemotherapy initiation; ^b^ Time from chemotherapy initiation until censoring at last contact; ^c^ Survival from the time of lymphoma diagnosis. Abbreviation: IQR: interquartile range.

**Table 1 cancers-17-01005-t001:** Demographic and clinical characteristics of participants.

Variable	Overall (*N* = 146)	Lymphoma (*N* = 21)	Non-Lymphoma (*N* = 125)	*p*-Value
**Age, years** (IQR)	40.0 (33.0–49)	46.0 (39.0–51)	39.0 (33.0–48)	0.1093 ^1^
**Gender** (%)				
Female	76/146 (52.1)	7/21 (33.3)	69/125 (55.2)	0.0635 ^2^
Male	70/146 (48.0)	14/21 (66.7)	56/125 (44.8)	
**Race** (%)				0.3510 ^2^
Black	141/146 (96.6)	21/21 (100.0)	120/125 (96.0)	
Colored	5/146 (3.4)	0/21 (0.0)	5/125 (4.0)	
**Wealth quintile** (%)				0.6160 ^2^
Quintile 1	19/97 (19.6)	0/9 (0.0)	19/88 (21.6)	
Quintile 2	20/97 (20.6)	2/9 (22.2)	18/88 (20.5)	
Quintile 3	20/97 (20.6)	2/9 (22.2)	18/88 (20.5)	
Quintile 4	22/97 (22.7)	3/9 (33.3)	19/88 (21.6)	
Quintile 5	16/97 (16.5)	2/9 (22.2)	14/88 (15.9)	
**Antiretroviral therapy** (%)				0.3247 ^2^
No	33/146 (22.6)	3/21 (14.3)	30/125 (24.0)	
Yes	113/146 (77.4)	18/21 (85.7)	95/125 (76.0)	
**How long on ARTs** (%)				0.0463 ^2^
<1 year	42/112 (37.5)	3/18 (16.7)	39/94 (41.5)	
>1 year	70/112 (62.5)	15/18 (83.3)	55/94 (58.5)	
**CD4 count** (IQR)	215.5 (102–408)	213.0 (137–329)	218.0 (83.0–431)	0.9188 ^1^
**Viral load** (%)				0.0845 ^2^
Detectable	85/146 (58.2)	13/21 (61.9)	72/125 (57.6)	
Undetectable	39/146 (26.7)	8/21 (38.1)	31/125 (24.8)	
Unknown	22/146 (15.1)	0/21 (0.0)	22/125 (17.6)	
**HIV VL, Log10**	2.66	2.66	2.65	0.9513 ^1^
(IQR; *n*)	(1.87–5.33; 85)	(2.08–4.86; 13)	(1.82–5.45; 72)	
**Prior history of TB** (%)				0.3190 ^2^
No	90/146 (61.6)	15/21 (71.4)	75/125 (60.0)	
Yes	56/146 (38.4)	6/21 (28.6)	50/125 (40.0)	
**Current treatment for TB** (%)				0.1745 ^2^
No	125/146 (85.6)	20/21 (95.2)	105/125 (84.0)	
Yes	21/146 (14.4)	1/21 (4.8)	20/125 (16.0)	
**ECOG Performance Status (%)**				0.3254 ^2^
≤1	116/146 (79.5)	15/21 (71.4)	101/125 (80.8)	
≥2	30/146 (20.6)	6/21 (28.6)	24/125 (19.2)	
**FNA Result** (%)				-
TB	61/146 (41.8)	0/21 (0.0)	61/125 (48.8)	
Lymphoma	18/146 (12.3)	18/21 (85.7)	0/125 (0.0)	
TB and lymphoma	1/146 (0.7)	1/21 (4.8)	0/125 (0.0)	
Reactive	17/146 (11.6)	2/21(9.5)	15/125 (12.0)	
Benign/cyst	16/146 (11.0)	0/21 (0.0)	16/125 (12.8)	
Other malignancy	8/146 (5.5)	0/21 (0.0)	8/125 (6.4)	
Inflammation/	12/146 (8.2)	0/21 (0.0)	12/125 9.6)	
abscess				
Inadequate	13/146 (8.9)	0/21 (0.0)	13/125 (10.4)	

For continuous variables, the median and IQR are shown. *N* is the number of non–missing values and *n* are the frequencies. Tests used: ^1^ Kruskal–Wallis; ^2^ Pearson’s Chi-squared test. Abbreviations: IQR: interquartile range; ECOG PS: Eastern Cooperative Oncology Group Performance Status; ART: antiretroviral therapy; VL: viral load; TB: tuberculosis.

**Table 2 cancers-17-01005-t002:** FNA result and diagnostic procedures.

	FNA Result	Other Procedures	Confirmatory Biopsy
**Biopsy confirmed**			
1	HL		Excisional LN biopsy—grey zone lymphoma (DLBCL/HL)
2	HL		Excisional LN biopsy—HL
3	Reactive	Repeat FNA showed DLBCL	Soft palate biopsy—DLBCL
4	HL		Neck biopsy—DLBCL
5	NHL	BMB—inadequate	Incisional LN biopsy—PBL
6	NHL	(1) LN core biopsy—inadequate (2) LN core biopsy—inadequate (3) Incisional LN biopsy—possible high-grade NHL	Excisional LN biopsy—DLBCL
7	NHL		LN core biopsy—B-cell NHL
8	DLBCL		Excisional LN biopsy—DLBCL
9	HL		LN core biopsy—HL
10	Reactive	Excisional LN biopsy—Castleman disease	BMB—HHV8+ DLBCL
11	NHL		LN biopsy—DLBCL
**Death prior to confirmation ^1^**			
12	NHL	LN core biopsy—non diagnostic	
13	NHL	BMB—non-diagnostic	
14	NHL		BMB—malignant infiltrate
15	NHL		BMB—DLBCL
16	HL		
17	NHL + TB ^2^		
18	NHL	(1) Excisional LN biopsy—non diagnostic (2) BMB—no infiltrate	LN core biopsy—Burkitt lymphoma
19	HL	Prior FNA showed HL	
**Alive without biopsy confirmation**			
20	HL		
21	NHL	Prior FNA and LN core biopsy—non diagnostic	

^1^ The eight participants who died prior to lymphoma confirmation are listed. Of note, three of these participants had a biopsy to confirm the diagnosis, but did not survive for the pathology to be finalized. ^2^ FNA cytology showed that NHL and GeneXpert were positive for TB. Abbreviations: FNA: fine needle aspiration; HL: Hodgkin lymphoma; LN: lymph node: DLBCL: diffuse large B-cell lymphoma; NHL: non-Hodgkin lymphoma; BMB: bone marrow biopsy; PBL: Plasmablastic lymphoma; HHV8: human herpesvirus 8; TB: tuberculosis.

**Table 3 cancers-17-01005-t003:** Participant intervals.

	Diagnosed with Lymphoma *N* = 11	Deceased Prior to Lymphoma Diagnosis *N* = 8	Alive, Awaiting Biopsy Confirmation *N* = 2
**Median patient interval in days** (*n*)	17 (7)	59 (7)	36 (1)
	(IQR: 11–71)	(IQR: 11–78)	(IQR: 36–36)
**Site of first presentation** (*n*)			
Clinic or hospital	64% (7)	25% (2)	50% (1)
Traditional healer	9% (1)	25% (2)	0% (0)
Pharmacy	27% (3)	50% (4)	50% (1)
**Median healthcare provider interval in days** (*n*)	85 (10)	97 (8) *	
	(IQR: 55–134)	(IQR: 33–134)	-
**Number treated with chemotherapy** (%)	9 (82%)	-	-
**Median number of days from presentation until chemotherapy** (*n*)	126 (9)		
	(IQR: 85–175)	-	-
**For participants not treated with chemotherapy, median number of days from presentation till death** (*n*)	75 (2)		
	(IQR: 71–80)	-	
**Number of deaths after chemotherapy initiated** (%)	3 (33%)	-	-
**Median survival/censored from presentation** (*n*)	467 (9)		277 (2)
	(IQR: 315–559)	-	(IQR: 232–323)

*N* is the number of non–missing values and *n* are the frequencies. * Healthcare provider interval for participants who died prior to a diagnosis represents the time from presentation to the health system until death.

## Data Availability

The deidentified data set, data dictionary, and study protocol are available upon request sent to the corresponding author (svogt2@jhmi.edu).

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
