# Peer review of "Prolonged Diagnostic Interval Leads to High Lymphoma Related Mortality in a Prospective Cohort of People with HIV Undergoing Fine Needle Aspiration"

_cancers, 2025, doi:10.3390/cancers17061005_

Round 1
Reviewer 1 Report
Comments and Suggestions for Authors
Broadly this is a very well written manuscript with adequate description of methods and results without overinterpretation of the data. This is an important and logical follow up to retrospective study from the same study group. The depth of the data here is remarkable and the authors should be commended for the transparency of all data around the lymphoma diagnoses that were made. Table 2 with data on the specific diagnoses and how they were made is incredibly helpful and important for the field. No predictive demographic data were identified for predicting lymphoma diagnosis in this patient population but reporting of that negative finding is important and useful. There may be more advanced analytic techniques that could be used to more definitively rule out those relationships but I agree are not necessary as there are no strong links by the eye test.
Minor comments:
Can you provide some context about where patients would be referred for lymph node biopsy, the process involved, and testing available for those patients?
Were patients with “inflammation” referred for further biopsy to rule out other diagnoses such as MCD?
Recommend renaming “patient interval” as this is not an intuitive label. It is labeled in the methods but might be better to use something else like “time from symptoms onset to first healthcare interaction” or something like that to improve clarity.
Author Response
Comment 1: Can you provide some context about where patients would be referred for lymph node biopsy, the process involved, and testing available for those patients?
Response 1: We thank the reviewer for this comment. We have included the statement below that describes the location of the hospital in Soweto and the general structure of the public healthcare system in Soweto. We also describe the referral process. The testing that is typically performed at the FNA clinic includes 2 passes. The first pass is sent for TB testing with GeneXpert Ultra and TB culture and the 2nd pass is sent for cytology review. We refer the reviewer to page 6 lines 141-143 where this is described.
“Geographically, Soweto is in the southwestern part of Johannesburg and has an estimated population of roughly 3 million, predominately black individuals. The public healthcare system in South Africa is a tiered system consisting of primary healthcare clinics (PHCs), community healthcare centers (CHCs), district, regional and tertiary hospitals. Patients presenting to the PHCs or CHCs that require a higher level of care, such as specialized diagnostic services, like an FNA, are referred to specialty clinics located at the referral hospitals. Chris Hani Baragwanath Academic Hospital has an established FNA clinic that accepts referrals from the 22 PHCs and six CHCs within Soweto.”
Comment 2: Were patients with “inflammation” referred for further biopsy to rule out other diagnoses such as MCD?
Response 2: In this observational study, the study team was not involved in the decision to refer patients for biopsy. Per the pathologist review of cases that fell under the inflammation category, clinical correlation was advised, however no formal recommendation for biopsy was noted. This is an important point that deserves closer scrutiny regarding the triage of patients with such a finding. As our study endpoint of interest was overall survival and lymphoma diagnosis, this was beyond the scope of the current study. Interestingly, we do note that one participant had an initial FNA showing reactive adenopathy (Participant 10, Table 2). They were then referred for lymph node biopsy and diagnosed with Castleman’s disease, followed by the diagnosis of an HHV8+ DLBCL on bone marrow biopsy. This sequence follows the practice of clinical correlation, whereby the treating clinical felt that further investigations were warranted based on the patient’s presentation.
Comment 3: Recommend renaming “patient interval” as this is not an intuitive label. It is labeled in the methods but might be better to use something else like “time from symptoms onset to first healthcare interaction” or something like that to improve clarity.
Response 3: We appreciate this comment as the defining and naming of these intervals has been the topic of discussion in the literature [1]. As a study team, we wanted to ensure that we defined the specific time intervals used in the methods to avoid confusion. We also wanted to ensure that our findings were consistent with other published reports, and specifically the report from Cape Town [2]. The Cape Town study used the terms, “Patient interval” and “Healthcare practitioner interval”. To ensure that our study measured similar time points, we thought it best to use the same terms. The patient interval has been defined in the Aarhus statement [1] and was included and defined in a published meta-analysis[3]. Our goal to use this same terminology was to decrease confusion and provide clarity within the context of the published literature on time to diagnosis. I agree that there is still work to be done in the field to ensure a universal definition.
- Weller D, Vedsted P, Rubin G, et al: The Aarhus statement: improving design and reporting of studies on early cancer diagnosis. Br J Cancer 106:1262-7, 2012
- Antel K, Levetan C, Mohamed Z, et al: The determinants and impact of diagnostic delay in lymphoma in a TB and HIV endemic setting. BMC Cancer 19:384, 2019
- Petrova D, Spacirova Z, Fernandez-Martinez NF, et al: The patient, diagnostic, and treatment intervals in adult patients with cancer from high- and lower-income countries: A systematic review and meta-analysis. PLoS Med 19:e1004110, 2022
Reviewer 2 Report
Comments and Suggestions for Authors
- It is intriguing that all patients were either black or coloured. Is this a selection byass? please clarify.
It is assumed that FNA diagnosis relies only in morphological evaluaion. Please explain whether ancillary techniques such as Flow citometry or immunocitochemistry were used or may potentially aid in the formal diagnosis of lymphoma.
Author Response
Comment 1: It is intriguing that all patients were either black or coloured. Is this a selection byass? please clarify.
Response 1: The study population is representative of the population that receives care at Chris Hani Baragwanath Hospital in Soweto, South Africa. Soweto has a population of roughly 3 million people. Per Stats SA, health statistics that are publicly available through the Department of Statistics in South Africa (https://www.statssa.gov.za/?page_id=4286&id=11317), the breakdown by race is 98.5% Black African and 1.0% Coloured in Soweto. We have added the following line to the methods section, “Geographically, Soweto is located in the southwestern part of Johannesburg and has an estimated population of roughly 3 million, predominately black individuals.”
Comment 2: It is assumed that FNA diagnosis relies only in morphological evaluaion. Please explain whether ancillary techniques such as Flow citometry or immunocitochemistry were used or may potentially aid in the formal diagnosis of lymphoma.
Response 2: This is a great point, and we thank reviewers for this comment. In the methods section we describe the testing that is routinely performed on FNA (page 6, lines 141-143). The testing that is typically performed at the FNA clinic includes 2 passes. The first pass is sent for TB testing with GeneXpert Ultra and TB culture and the 2nd pass is sent for cytology review.
Currently Flow cytometry and IHC are not routinely performed on the FNA samples. We suspect that the high burden of TB partially explains why they are not performed and, in many cases, would not be necessary. In the discussion we also make reference to the sheer volume of FNAs that are performed within the public healthcare sector in Johannesburg. This was noted in our prior retrospective study. The decision to incorporate upfront Flow or IHC would need to be made at the public health system level and would likely need to a formative economic analysis to see if it is economically feasible in light of the majority of cases where it would not provide benefit.
Furthermore, one could argue that FNA cytology does a good job at identifying patients harbouring an aggressive lymphoma. The issue we have identified in the current system, is the poor linkage to a diagnostic biopsy. Flow cytometry and IHC could be useful for those harbouring a lymphoma, but to have an impact they would need to provide a diagnosis in the absence of a diagnostic biopsy, otherwise linkage to biopsy would still be a problem. As FNA alone, without a core-biopsy is the standard of care, it would be difficult to arrive at a definitive diagnosis with classification even with Flow and IHC. Many studies still recommend formal biopsy to confirm abnormal results by Flow or IHC, and some lymphomas can be missed using these tests (like Hodgkin lymphoma), therefore a negative result would not rule out the need for a formal biopsy[4,5]. Logistically, for these tests to be useful, they would need to be performed upfront on all patients, and a negative result would not rule out a lymphoma.
While the addition of these tests to the current diagnostic paradigm is beyond the scope of the current observational study, we have added the following statement to the discussion, “The addition of upfront Flow cytometry and/or immunohistochemistry to the FNA procedure could be considered. This strategy would come at a significant cost to the health system and would not eliminate the need for confirmatory biopsy in many cases.”
- Dunphy CH: Applications of flow cytometry and immunohistochemistry to diagnostic hematopathology. Arch Pathol Lab Med 128:1004-22, 2004
- Swart GJ, Wright C, Brundyn K, et al: Fine needle aspiration biopsy and flow cytometry in the diagnosis of lymphoma. Transfus Apher Sci 37:71-9, 2007
Reviewer 3 Report
Comments and Suggestions for Authors
Reviewer Comments
The manuscript titled "Prolonged Diagnostic Interval Leads to High Lymphoma Related Mortality in a Prospective Cohort of People with HIV Undergoing FNA," authored by Samantha L. Vogt, Khuthadzo Hlongwane, Arshia Arora, Kennedy Otwombe, Deshan Chetty, Rebecca H. Berhanu, Ziyaad Waja, Wendy Stevens, Tanvier Omar, Neil A. Martinson, Richard F. Ambinder and Rena R. Xian, presents valuable insights into the diagnostic challenges and outcomes of HIV-positive patients undergoing fine needle aspiration (FNA) for lymphadenopathy in South Africa.
However, the manuscript requires revisions based on the following suggestions:
Comments
- The small sample size (21 lymphoma cases) limits statistical power; please clarify how the sample size was determined and consider adding confidence intervals for key measurements.
- The wealth quintile analysis may be underpowered due to the small number of samples in each category.
- Provide more details on the follow-up protocol for patients awaiting biopsy and explain how missing data was handled in the analysis.
- The definition of "healthcare provider interval" could be more precise and also consider providing more context for the structure of the local healthcare system.
- More details are needed on how recall bias was addressed beyond electronic database verification.
- Table 2 could be simplified to improve readability.
- The mortality analysis could benefit from Kaplan-Meier survival curves.
- Consider analyzing factors associated with longer diagnostic delays.
- The limitations section could be expanded.
- More emphasis is needed on practical recommendations for improving the diagnostic pathway.
- There are some inconsistencies in number formatting throughout the tables, e.g., Viral load (%) Detectable 85/146 (58.2), and HIV VL, Log10 (IQR) 2.66 (1.87-5.33).
Author Response
Comment 1: The small sample size (21 lymphoma cases) limits statistical power; please clarify how the sample size was determined and consider adding confidence intervals for key measurements.
Response 1: This study was designed as an observation study and the descriptive results are reported. This study was not powered to detect differences between those that were diagnosed with lymphoma and those that were not, nor those that survived till diagnosis and those that died. Given the relatively modest number of lymphomas, we did not want to “overinterpret our data”, as was noted by reviewer one. We agree that this is an important limitation to our current study and have added this to the limitation section in the discussion.
Continuous variables, including the diagnostic intervals are all reported in medians and have associated interquartile ranges provided. The interquartile range allows for the effect of outliers to be displayed.
Comment 2: The wealth quintile analysis may be underpowered due to the small number of samples in each category.
Response 2: As above, we acknowledge that a limitation of our current study is a lack of power to detect significant differences between these groups. This was not the primary or secondary objective of this descriptive study.
Comment 3: Provide more details on the follow-up protocol for patients awaiting biopsy and explain how missing data was handled in the analysis.
Response 3: All study participants underwent a uniform study follow-up that is described in the methods section.
“Study participants were prospectively followed for at least eight months after the FNA procedure with serial follow-up visits scheduled at 2, 6, 14, 24, and 32 weeks. At each follow-up visit, participants were asked whether they had received the results of the FNA, any further work-up that they had undergone including referral for biopsy (if indicated), and a brief clinical history, symptom assessment and physical exam were performed.”
Additionally, we have added the following text to the manuscript, “At study follow-up visits, participants that had not received their FNA results were provided a printed copy of the result, provided the result was available in the NHLS system. All participants were encouraged to return to their clinic for follow-up to receive/discuss their FNA results and any ongoing symptoms that they were experiencing.”
For missing data, on Table 1, we note varying degrees of missing data for the demographic and clinical characteristics. The denominator has been provided for all variables to provide transparency in the degree of missing data. For the study follow-up visits, our outcomes of interest were a diagnosis of lymphoma and survival at 8 months. The following statement was added to the methods, “Participants were prospectively followed until a diagnosis of lymphoma was established. A secondary outcome was 8-month survival.” In our results section, page 14, line 298 – 299 were report on 2 patients that were lost to follow-up. Individual missed visits did not impact on the study outcomes of interest. As per above, we conducted an observational study and did not refer participants for procedures/biopsy.
Comment 4: The definition of "healthcare provider interval" could be more precise and also consider providing more context for the structure of the local healthcare system.
Response 4: We appreciate this comment as the defining and naming of these intervals has been the topic of discussion in the literature[1]. As a study team, we wanted to ensure that we defined the specific time intervals used in the methods to avoid confusion. For the healthcare provider interval, we have clarified that time of presentation to the healthcare system represents the time a participant presented to a clinic or hospital, until a lymphoma diagnosis was confirmed on biopsy (date of biopsy result).
- Weller D, Vedsted P, Rubin G, et al: The Aarhus statement: improving design and reporting of studies on early cancer diagnosis. Br J Cancer 106:1262-7, 2012
We have also included the statement below that describes the location of the hospital in Soweto and the general structure of the public healthcare system in Soweto.
“Geographically, Soweto is located in the southwestern part of Johannesburg and has an estimated population of roughly 3 million, predominately black individuals. The public healthcare system in South Africa is a tiered system consisting of primary healthcare clinics (PHCs), community healthcare centers (CHCs), district, regional and tertiary hospitals. Patients presenting to the PHCs or CHCs that require a higher level of care, such as specialized diagnostic services, like an FNA, are referred to specialty clinics located at the referral hospitals. Chris Hani Baragwanath Academic Hospital has an established FNA clinic that accepts referrals from the 22 PHCs and six CHCs within Soweto.”
Comment 5: More details are needed on how recall bias was addressed beyond electronic database verification.
Response 5: At the time of enrolment, which occurred on the same day as the FNA procedure, all participants were asked the same question, “When did your symptoms start” and “When did you first seek care”. Uniformly collecting this data at time of enrolment was utilized the reduce the recall period. The other strategy that we utilized was verifying the date of presentation to an objective data source. In the public healthcare sector in South Africa, electronic medical records for clinic visits and notes are not available. The only objective data source that is available is the electronic laboratory database. We note recall bias as a limitation in our study and have reworded the limitations section of the discussion as follows,
“To limit the effect of recall bias, we utilized an electronic laboratory database, an objective data source, to help corroborate dates reported by participants when possible. Additionally, the date of symptom onset and presentation to the healthcare system were uniformly collected at time of enrollment for all study participants to decrease the recall period.”
Comment 6: Table 2 could be simplified to improve readability.
Response 6: We appreciate this comment. As the results of our study are primarily descriptive, the specific diagnostic procedures that participants underwent show the complexity of the diagnostic process for lymphoma. Given our focus on the healthcare provider interval, it is important to describe the procedures that were occurring during this interval. We have modified the table as there were some inconsistencies with how the biopsies were reported. The additional detail regarding core vs incisional vs excisional was added when known.
Comment 7: The mortality analysis could benefit from Kaplan-Meier survival curves.
Response 7: As previously mentioned, we were cautious in overinterpreting our findings given the sample size. As we were not powered to detect differences between groups, the Kaplan-Meier curves would be difficult to interpret.
Comment 8: Consider analyzing factors associated with longer diagnostic delays.
Response 8: Similar to above, this study was not powered to detect differences between groups. As was pointed out, with only 21 participants in the lymphoma category, modelling predictors of diagnostic delay would be difficult to interpret.
Comment 9: The limitations section could be expanded.
Response 9: We thank the reviewer for this comment and have added the following to the limitations section, “As this study was designed as an observational study, and a relatively modest number or participants (n=21) were included in the lymphoma category, our results lack the power to detect differences between those with lymphoma and those without, and those that survived to a lymphoma diagnosis and those that died.”
Comment 10: More emphasis is needed on practical recommendations for improving the diagnostic pathway.
Response 10: We have expanded this discussion to include the addition of Flow cytometry or immunohistochemistry to the FNA procedure and have included a potential for a care navigator.
“The addition of upfront Flow cytometry or immunohistochemistry to the FNA procedure could be considered. This strategy would come at a significant cost to the health system and would not eliminate the need for confirmatory biopsy in many cases. A more practical approach could include the addition of a care-navigator to facilitate the linkage between the FNA and diagnostic biopsy.”
Comment 11: There are some inconsistencies in number formatting throughout the tables, e.g., Viral load (%) Detectable 85/146 (58.2), and HIV VL, Log10 (IQR) 2.66 (1.87-5.33).
Response 11: Thank you for this comment we have added the (n) to the HIV VL variable to provide clarity.
Round 2
Reviewer 3 Report
Comments and Suggestions for Authors
Thank you for your detailed response to my review comments. I appreciate your time and effort in addressing each question thoroughly.
Best regards,